

# A century of ecosystem change: human and seabird impacts on plant species extirpation and invasion on islands

Thomas K. Lameris[1,2,3], Joseph R. Bennett[1,4], Louise K. Blight[1,5], Marissa Giesen[1,2], Michael H. Janssen[6], Joop J.H.J. Schaminée[2] and Peter Arcese[1]

[1] Department of Forest and Conservation Sciences, University of British Columbia, Vancouver, British Columbia, Canada
[2] Plant Ecology and Nature Conservation Group, Wageningen University, Wageningen, The Netherlands
[3] Department of Animal Ecology, Netherlands Institute of Ecology, Wageningen, The Netherlands
[4] Institute of Environmental Science and Department of Biology, Carleton University, Ottawa, Ontario, Canada
[5] Procellaria Research & Consulting, Victoria, British Columbia, Canada
[6] Wildlife Research Division, Environment and Climate Change Canada, Ottawa, Ontario, Canada

Corresponding author
Thomas K. Lameris,
thomaslameris@gmail.com

## ABSTRACT

We used 116 years of floral and faunal records from Mandarte Island, British Columbia, Canada, to estimate the indirect effects of humans on plant communities via their effects on the population size of a surface-nesting, colonial seabird, the Glaucous-winged gull (*Larus glaucescens*). Comparing current to historical records revealed 18 extirpations of native plant species (32% of species historically present), 31 exotic species introductions, and one case of exotic introduction followed by extirpation. Contemporary surveys indicated that native species cover declined dramatically from 1986 to 2006, coincident with the extirpation of 'old-growth' conifers. Because vegetation change co-occurred with an increasing gull population locally and regionally, we tested several predictions from the hypothesis that the presence and activities of seabirds help to explain those changes. Specifically, we predicted that on Mandarte and nearby islands with gull colonies, we should observe higher nutrient loading and exotic plant species richness and cover than on nearby islands without gull colonies, as a consequence of competitive dominance in species adapted to high soil nitrogen and trampling. As predicted, we found that native plant species cover and richness were lower, and exotic species cover and richness higher, on islands with versus without gull colonies. In addition, we found that soil carbon and nitrogen on islands with nesting gulls were positively related to soil depth and exotic species richness and cover across plots and islands. Our results support earlier suggestions that nesting seabirds can drive rapid change in insular plant communities by increasing nutrients and disturbing vegetation, and that human activities that affect seabird abundance may therefore indirectly affect plant community composition on islands with seabird colonies.

## INTRODUCTION

Species invasion and long-term change in communities are key themes in ecology (*Pickett, Collins & Armesto, 1987*; *Strayer et al., 2006*) but rarely studied over multiple decades (*Bakker et al., 1996*). Comparing contemporary and historical surveys can help rectify this deficit and enhance understanding of long-term ecological change (*Macdougall & Turkington, 2005*; *Arcese et al., 2014*; *McKechnie et al., 2014*). In particular, human disturbance, habitat conversion and exotic species invasion are widely acknowledged drivers of plant community change (*Davis, Grime & Thompson, 2000*; *Macdougall & Turkington, 2005*; *Seabloom et al., 2006*) and can act indirectly following predator removal or herbivore introduction to facilitate trophic cascades, particularly on islands (*Estes et al., 2011*; *Arcese et al., 2014*). Similar observations suggest that humans may also affect island plant communities indirectly by affecting population size in island-nesting seabirds, such as by introducing predators, harvesting, providing anthropogenic foods or depleting their prey (*Croll et al., 2005*; *Mulder et al., 2011*; *Baumberger et al., 2012*). Physical disturbance and chemical inputs to vegetation have often been linked to changes in nest density or guano deposition by gulls (Laridae; *Sobey & Kenworthy 1979*; *Ellis, 2005*) and cormorants (Phalacrocoracidae; *Ishida, 1996*; *Ishida, 1997*). On islands with historically low seabird abundances, increased guano deposition may cause long-lasting changes in soil chemistry and nutrients (*García et al., 2002*; *Wait, Aubrey & Anderson, 2005*; *Caut et al., 2012*) that facilitate increases in the cover and richness of species adapted to nutrient-rich soils (*Baumberger et al., 2012*), but reducing cover and richness in species adapted to poor, shallow soils (*García et al., 2002*; *Ellis, 2005*). We therefore expected that increasing seabird populations and guano deposition on islands will favour some plant species over others and drive plant community composition (*Baumberger et al., 2012*).

In this paper, we provide a case study to illustrate these ideas by using multiple data sources to examine the effects of seabirds on plant communities on islands of the Georgia Basin, British Columbia (BC), Canada. We focus on Mandarte Island, which has been surveyed sporadically for vegetation cover and regularly for the abundance of nesting glaucous-winged gulls (*Larus glaucescens*). Gulls increased rapidly after gaining protection in the early 1900s, when about 240 pairs nested on Mandarte, to 2,500 breeding pairs in 1985 (*Blight, Drever & Arcese, 2015*, Supplementary Material), consistent with the finding that human activities have been the key driver of change in the abundance of colonially nesting gulls in our study region over the last century (*Blight et al., 2014*; *Blight, Drever & Arcese, 2015*; *Hobson, Blight & Arcese, 2015*). The number of nesting cormorants (*Phalacrocorax* spp.) on Mandarte Island also increased to the mid-1980s following their protection (*Drent et al., 1964*; *Chatwin, Mather & Giesbrecht, 2002*).

We asked whether these impacts of humans on regional seabird populations may also have contributed indirectly to long-term change in plant communities on Mandarte Island. To do so, we used opportunistic and systematic data sets collected over 116 years

(non-continuously), including historical photographs, expedition notes, published accounts, and historical and contemporary surveys of plant species cover, richness and soil nutrients to ask: (1) how native and exotic plant species richness and cover changed on Mandarte Island from 1896 to 2012; and (2) what contemporary relationships exist between gull presence and plant community composition on islands in our study region with and without gull breeding colonies. Our results support the suggestion that humans affect plant communities indirectly via their effects on seabird abundance (*Croll et al., 2005*) and imply that population increases in gulls over the last century have facilitated rapid change in plant communities on islands with nesting seabirds.

## MATERIALS AND METHODS

### Study site

Contemporary surveys of plant communities occurred on Mandarte (48.38 N, 123.17 W; 6.8 ha) and 24 nearby islands (0.3–11.5 ha, within a 100 km radius of Mandarte). All islands are uninhabited, experience a sub-Mediterranean climate regime of mild, wet winters and warm, dry summers (*MacDougall et al., 2006*), and support maritime meadow and Garry oak (*Quercus garryana*) ecosystem flora. This ecosystem has declined 95% in extent in Canada since 1860 and is now severely threatened in the Georgia Basin region (*Gedalof, Pellat & Smith, 2006*; *Lea, 2006*; *Bennett & Arcese, 2013*). Mandarte Island supports a mix of mature shrubs on deeper soils and herbaceous meadows in shallower soils, and was managed for aboriginal plant food harvest prior to European colonization of the region (e.g., *Lea, 2006*; *Arcese et al., 2014*). Mandarte Island also hosts the largest seabird colony in South-Western BC, with, over time, an estimated 200–1650 breeding pairs of pelagic (*P. pelagicus*) and double-crested cormorants (*P. auritus*), 100–150 pairs of pigeon guillemots (*Cepphus columba*) and 5–125 pairs of rhinoceros auklets (*Cerorhinca monocerata*), all nesting on cliff ledges (cormorants) or in burrows (*Chatwin, Mather & Giesbrecht, 2002*, P. Arcese, 2016, unpublished data). Glaucous-winged gulls (*Larus glaucescens*) nest mainly in the island's open or shrubby meadows and have varied in number from c 240 to 2,500 breeding pairs, with a peak count in 1985 (*Drent & Guiguet, 1961*; *Drent et al., 1964*; *Campbell et al., 1990*; *Blight, 2012*; *Blight, Drever & Arcese, 2015*), but also on other islands throughout the Georgia Basin (*Vermeer & Butler, 1989*).

Mandarte is similar in size and isolation to many islands that we surveyed without seabird colonies, all of which are likely to have been visited regularly by humans in the last century. From the 1860s–1900s, most islands were probably visited by colonist-explorers and First Nations and colonist foragers. Pleasure boaters visited and camped on most of the islands we surveyed until 2003, when many were protected from human access. Although we cannot estimate human visitation rate to the islands we surveyed, we would not expect that differences in human visitation could account for the vegetation communities we surveyed on Mandarte and other islands in the vicinity.

### Historical records from Mandarte Island

Partial accounts of Mandarte Island flora and fauna appear as museum records as early as 1896, including notes on seabird presence and abundance, plant species occurrence,

and photographs, all depicting a vegetation community historically characteristic of a maritime Garry oak ecosystem (e.g., *Tompa, 1963*; *Drent et al., 1964*). Various researchers subsequently conducted periodic surveys after 1955, mainly recording plants as species lists and seabirds as counts, but also documenting early changes in the plant community and linking those changes to contemporary increases in gull and cormorant numbers (*Drent et al., 1964*). We compiled historical records of plant species occurrence from the archived museum reports, herbarium specimens and lists provided by *Tompa (1963)* and *Drent et al. (1964)*, and then compared these lists to contemporary surveys conducted by P.A., M.G. and T.K.L. Here, we list all plant species recorded on Mandarte Island to 2012 and classified each as extirpated or extant, and as native or exotic to the Pacific coast of North America (Table S1). Historical photos of the island also allow qualitative comparisons to a 1963 photo by P. Grant covering a sizeable portion of Mandarte Island and including all typical habitats, which we replicated in 2002 (Fig. 1).

## Vegetation surveys

To estimate temporal change in shrub species richness and cover on Mandarte Island, we surveyed vegetation cover from June–July, 1986 and 2006, in 132, 20 × 20 m grid-squares mapped onto a 1974 air photo. Surveys comprised c 90% of island area, but excluded some areas at the edge of sparsely- or non-vegetated bluffs. In each square we estimated the percent cover of all perennial shrubs and common herbaceous plants by species, exposed rock, and all graminoids as a group. Soil depth was estimated by hammering a graduated rod into the soil at five equidistant locations at the square and 1 m in from each corner of the square.

In June 2012, we also conducted contemporary surveys on islands adjacent to Mandarte Island to compare native species richness and cover, exotic species richness and cover, and total N and C in soil on islands with and without nesting glaucous-winged gulls, including three islands with gull colonies ('gull islands'), namely Mandarte, Arbutus (48.71 N, −123.44 W, 35–150 nesting pairs, 1976–2010) and the unnamed islet south of Mandarte (48.63 N, −123.28 W, 0–106 nesting pairs, 1962–2010; *Drent et al., 1964*; *Blight, 2012*). Cover of herbaceous and grass species and bare soil were estimated following *Bennett et al. (2012*; N = 14 × 1 m$^2$ quadrats on Mandarte Island; N = 4 on Arbutus Island; N = 3 on unnamed islet*). Quadrat locations were selected randomly within grid cells, for a random set of all mapped grid-squares that included meadow habitat. Plot-level data on soil depth was collated as the mean of four depths at quadrat corners. Total soil nitrogen, carbon and moisture were estimated by collecting 100 g of soil from five sub-samples ca. 2 cm below the rooting zone, then sieved (2 mm) and ground (≤ 0.14 mm) for analysis (Fisons NA-1500 combustion elemental analyser). Soil moisture was estimated from oven-dried (105 °C) sub-samples. Data from 21 quadrats collected on the three gull islands were compared to 57 quadrats from 22 islands in the same region not known to have supported persisting gull populations ('non-gull islands'; Table S2).

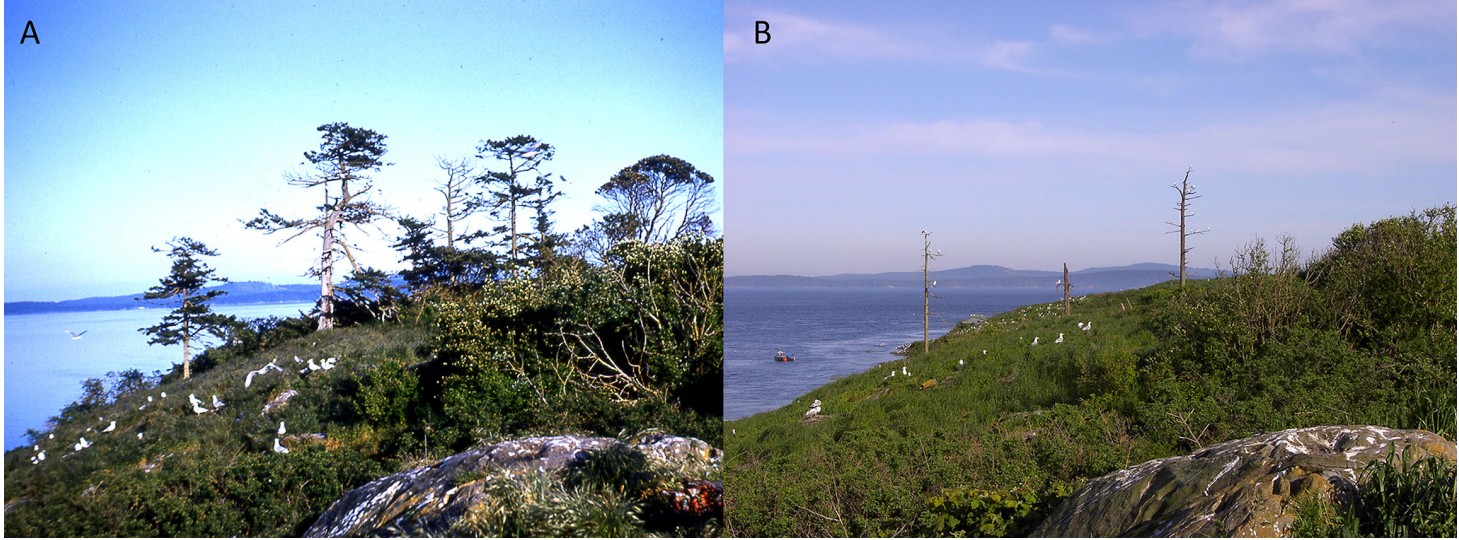

**Figure 1 Comparison of photographs of central woodland on Mandarte Island (1963–2002).** (A) Historical photograph from 1963 (by Peter Grant; *Smith et al., 2006*) and (B) a recent reproduction made in 2002 (by Peter Arcese). Note mature *Arbutus menziesii* and *Pseudotsuga menziesii* in the historical photograph. The latter include 3 individuals 0.8–1.4 m in basal diameter, and estimated at 250–400 years-old by comparison to newly harvested trees on adjacent islands.

For permission to access study sites, we obtained permits from Parks Canada and the Gulf Islands National Park Reserve (permit number #GINP-2008-1433), the US Fish and Wildlife Service (permit number #07002) and were given oral permission by Tswaout and Tseycum bands and by A.J. Brumbaum and P. Green.

## Statistical analyses

Vegetation survey data from Mandarte Island collected in paired plots in 1986 and 2006 were compared using Wilcoxon matched-pairs tests. For comparisons of quadrat data on gull and non-gull islands we used Wilcoxon signed-rank tests to test for differences in native and exotic species richness and cover and total soil N and C. To test for links between native and exotic species richness, cover, and soil conditions on gull versus non-gull islands we used generalized linear mixed models. Specifically, we constructed models to explain cover and richness, native cover and richness, and the proportion of exotic vs. native cover and richness using soil depth, total soil N and C, and presence of seabirds as fixed effects. We included island identity as a random factor in these models to account for repeat sampling within islands. Percentage cover was transformed using arcsine square root (for proportion) and richness was transformed using log+1 to normalize residual errors. We selected models based on AIC and estimated p-values for predictors using Markov-chain Monte Carlo sampling (*Baayen, Davidson & Bates, 2008*). Total N and C were highly correlated; however, because exploratory models indicated that each variable explained an independent fraction of variation in plant survey data, both variables were retained in our analyses, but interpreted cautiously. All mixed models used the lme4 package (*Bates, Maechler & Bolker, 2012*) in R v.2.15.1 (*R Development Core Team, 2014*).

## RESULTS

### Historical comparisons

Over 116 years we detected 18 extirpations of native species from Mandarte Island, representing a loss of 32% of all native plant species recorded there. In contrast, we detected 31 colonization events by exotic species, two apparent colonisations of native species, and four extirpations of exotic species known to have become established prior to 1960 (Table S1). Forty-five native and one exotic species identified on Mandarte Island prior to 1960 remained extant on the island as of 2012, but 94% of exotic colonization events occurred after 1960. Visual comparison of historic and contemporary photos (Fig. 1) and our observations over 35 years (PA) also suggest long-term declines in shrub cover and extent of bare rock. Photos document the extirpation of three tree species: grand fir (*Abies grandis*), arbutus (*Arbutus menziesii*) and Douglas-fir (*Pseudotsuga menziesii*). Comparing the diameter of dead Douglas-fir on Mandarte (c 60–180 cm DBH) to felled trees of similar size on adjacent islands suggests the largest individuals on Mandarte Island recruited about 300 yrs BP (P. Arcese, 2016, unpublished data).

### Contemporary surveys

Comparisons of contemporary surveys in 1986 and 2006 indicate a 31% decline in shrub cover on Mandarte Island (N = 132, p = 0.002). On average, snowberry (*Symphoricarpos albus*) declined from 33 to 19% cover (p < 0.001), Nootka rose (*Rosa nutkana*) from 24 to 20% (p = 0.04), and gooseberry (*Ribes divericatum*), from 3 to 0.8% (p < 0.001), whereas the exotic Himalayan blackberry (*Rubus armeniacus*) and native red elderberry (*Sambucus racemosa*) increased from 2 to 18% and 1 to 6%, respectively (p < 0.001). The extent of bare rock declined from 38 to 34% (p = 0.03) as grass cover increased (31 to 40%; p < 0.001).

### Influence of gull colonies

Surveys of island meadow-plants indicate that exotic species cover was significantly higher on islands with (63%, N = 21) versus without gull colonies (12%, N = 57; Fig. 2). In contrast, native species cover and richness were lower on islands with versus without gull colonies (cover: 28.7 ± 6.1% with vs. 45.9 ± 3.2% without; p = 0.01; richness: 1.67 ± 0.30 species with versus 7.30 ± 0.49 species without; p < 0.001; Fig. 2).

Quadrats on gull islands also yielded higher total nitrogen (N) and total carbon (C) concentrations than on non-gull islands (nitrogen: 2.86 ± 0.17% with vs. 1.66 ± 0.10% without; p < 0.001; carbon: 30.31 ± 2.03% with vs. 23.19 ± 1.44% without; p = 0.01). Total C and N were also positively related to soil depth ($r$ = 0.58, p = 0.006 and $r$ = 0.51, p = 0.02; respectively), but only on gull-islands. In contrast, mean soil depth was similar on gull and non-gull islands (11.14 ± 1.33 with vs. 19.37 ± 2.51 cm).

The presence of gull colonies was the only fixed effect retained in generalized mixed models to predict exotic cover (positive effect), the proportion of exotic versus native cover (positive effect), and native richness (negative effect; Table 1). In contrast, total C was the only fixed effect retained in models predicting exotic richness (negative effect) and

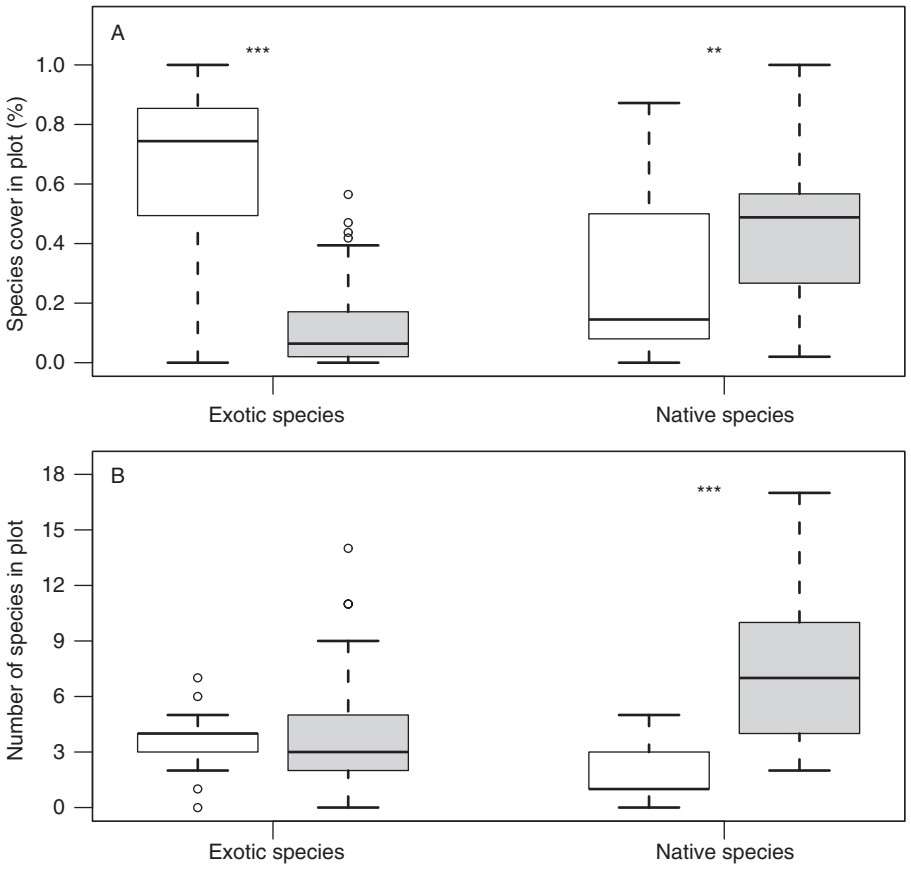

**Figure 2 Native and exotic species cover and richness on islands with and without gull colonies.** Box-plots showing differences in (A) species cover and (B) species richness of exotic and native species, on islands with gull colonies (white columns) and without (grey columns). Asterisks denote significant differences in Wilcoxon tests (double: p < 0.01; triple: p < 0.001).

**Table 1 Significant coefficients from several generalized mixed models relating vegetation composition and environmental variables.**

| Response variable | Predictor variables | Coefficients | Coefficient value | Standard error | t | p |
|---|---|---|---|---|---|---|
| Proportion exotic cover[1] | N, C, S, B | B | 0.449 | 0.162 | 2.776 | 0.007 |
| Exotic cover[1] | N, C, S, B | B | 0.541 | 0.123 | 4.417 | < 0.001 |
| Proportion exotic richness[1] | N, C, S, B | C, N | −0.021, −0.222 | 0.007, −0.092 | −2.964, −2.416 | 0.004, −0.018 |
| Exotic richness[2] | N, C, S, B | C | −0.008 | 0.003 | −2.762 | 0.007 |
| Native cover[1] | N, C, S, B | C | 0.008 | 0.003 | 2.225 | 0.029 |
| Native richness[2] | N, C, S, B | B | −0.496 | 0.056 | −8.884 | < 0.001 |

**Notes:**
Response variables:
[1] Transformed using arcsine square root.
[2] Transformed using log (+1).
Predictor variables: N, total nitrogen concentration; C, total carbon concentration; S, log mean soil depth; B, presence of gull colonies on island (with or without gull colonies).

native cover (positive effect). Both total C and total N were retained in the top model to predict proportion exotic and native species richness, whereas soil depth was not retained in any top model (Table 1).

## DISCUSSION

A comparison of historical and modern vegetation surveys confirms that substantial change in plant species cover, richness and origin occurred from 1896–2012 on Mandarte Island. Observations from historic photographs, plant lists and anecdotal reports are also consistent with the idea that these changes were driven in part by increases in the abundance of nesting seabirds on the island after 1920. Contemporary surveys of 24 islands with and without nesting seabirds further suggested that nesting seabirds increase soil C, N and depth, which can facilitate non-native grasses and inhibit recruitment and survival in native plant species adapted to poor, shallow soils (e.g., *Bennett et al., 2012*; *Best & Arcese, 2009*). Gulls and other surface-nesting seabirds such as cormorants are also reported as agents of change in soil and plant communities elsewhere (*Ellis, 2005*; *Mulder et al., 2011*). Thus, despite limits on the quality and scope of our data, our results are consistent with the idea that long-term vegetation change on Mandarte Island has been caused in part via the effects of humans on seabird abundance.

Direct impacts of seabirds on vegetation include the input and distribution of guano, which can be toxic or inhibit photosynthesis (*Ishida, 1997*), and physical disturbance due to trampling, nest construction and territory defence (*Sobey & Kenworthy, 1979*; *Ellis, 2005*). As reported elsewhere, we also found that islands with gull colonies had higher soil C and N concentrations than those without them (e.g., *Anderson & Polis, 1999*; *Wait, Aubrey & Anderson, 2005*). *Ellis (2005)* suggested that surface-nesting seabirds enhanced soil depth in systems without mammalian herbivores by facilitating litter accumulation; this observation is consistent with an observed decline in exposed rock on Mandarte Island, and with positive correlations between soil depth and C and N concentrations on islands with (but not without) gull colonies. In addition to the effects of seabirds, however, we also acknowledge that many other changes have occurred in our study system over the last century, including changes in the use of fire to maintain communities' aboriginal peoples of the region (*Macdougall & Turkington, 2005*, *Arcese et al., 2014*), as well as increased visitation by European colonists and others. While it is possible that the cessation of burning might increase soil depth over time, burning also ceased on a number of the islands without nesting seabirds, which still support most or all of the plant species extirpated from Mandarte Island (*Bennett & Arcese, 2013*). We are also unaware of any systematic differences between the islands we surveyed with and without gulls linked to their size, isolation or human visitation rates that could account for the floristic differences we report.

### Plant species cover and richness

A key result of our work was the observation that 32% of 63 native plant species historically extant on Mandarte Island were extirpated by 2012. In contrast, we documented 31 colonizations of Mandarte Island by exotic plant species. Although we lack precise dates for introductions, many appear to have been facilitated by exotic birds, such European starling *Sturnus vulgaris,* which began using Mandarte Island as a night roost in the 1970s and are well-documented as seed dispersers (*Arcese, 1989*; *McAtee, 2009*; *Bennett et al., 2011*). However, given that many exotic plant species that we detected are

widely distributed in the Georgia Basin and good dispersers (*Bennett, 2014*), it is also possible that some invasions we detected were facilitated by human visitors. Exotic grasses abundant on Mandarte Island, such as *Bromus rigidus* (19.6% cover) and *Dactylis glomerata* (14.5% cover), also occur in many Garry oak and maritime meadows, but are only common in those disturbed by humans or livestock (e.g., *Gonzales & Arcese, 2008*; *Bennett et al., 2012*), especially in nutrient-rich soils (*Klinkenberg, 2012*). These points suggest that the invasion, establishment and dominance of at least some exotic invaders to Mandarte Island was facilitated by interactions between birds, humans and changing soils.

Three tree species established on Mandarte Island prior to 1896 remained extant into the 1960s (arbutus), 1980s (grand fir) and 1990s (Douglas-fir), but were extirpated by 2002, including individual Douglas-firs likely to have been $\geq$ 300 years-old (Fig. 1). Although declines in conifer survival have been linked to decreased precipitation in our study region (*Murdock et al., 2012*), guano and physical damage from perching cormorants is well-known to kill trees in other systems (*Hebert et al., 2005*; *Boutin et al., 2011*) and was reported to be underway on Mandarte Island about 1960 (*Drent et al., 1964*). Declines in shrub cover on Mandarte Island may also be due in part to guano deposition, although the trampling and shredding of shrubs and adventitious shoots by gulls was widely evident after 1981 (P. Arcese, 2016, unpublished data). The accumulation of soil N and C and promotion of grass cover at the expense of shrub recruitment is also documented from seabird colonies elsewhere (*Ishida, 1996*; *Ishida, 1997*; *Ellis, 2005*). The colonization and spread of exotic blackberry and native red elderberry may also contribute to reduced native shrub cover, as these species are adapted to shallow soils and summer drought, whereas the aforementioned species favor the moist, nutrient-rich soils which have developed more recently on Mandarte Island (e.g., *Bennett et al., 2012*). Overall, these observations are also consistent with the idea that increases in seabird abundance have contributed directly or indirectly to declines in native shrub cover on Mandarte Island via a combination of mechanisms described above.

## Plant communities and gull colonies

Our observation that exotic species cover was higher on islands with than without gull colonies is consistent with our finding of enriched total C and N on islands with gulls, and with earlier suggestions that increases in exotic species cover co-occurred with increases in gull and cormorant abundance on islands (*Hogg & Morton, 1983*; *Ellis, 2005*; *Baumberger et al., 2012*). In addition, *Best & Arcese (2009)* showed experimentally that trampling by exotic Canada geese (*Branta canadensis*) reduced native species richness and increased exotic species dominance in maritime meadows in the Georgia Basin, confirming that other species can also influence plant community change in this system, particularly on small islets (*Best & Arcese, 2009*; *Isaac-Renton et al., 2010*; *Bennett & Arcese, 2013*).

The effects of seabirds on island plant communities in our study region are less certain into the future, given that gulls and cormorants have declined from peak abundances estimated in the mid-1980s. In gulls, reductions in diet quality linked to human overfishing and the substitution of anthropogenic for marine foods, perhaps aided by the

recovery of bald eagle numbers, may all have acted to reduce their abundance, including on Mandarte Island (*Blight, Drever & Arcese, 2015*). Declines in cormorant populations have also occurred but remain unexplained (*Chatwin, Mather & Giesbrecht, 2002*).

It remains to be seen whether these changes will result in the recovery of native dominance if disturbance is sufficiently reduced and nutrient profiles attain the levels indicated on islands without nesting seabirds. Nevertheless, our results offer a baseline for comparisons in future, particularly of islands in the region with and without nesting seabirds present.

## ACKNOWLEDGEMENTS

We thank N. Turner for extensive discussion, P. Grant for photographs, W. Cornwell, the Arcese lab and NCP chair group for feedback, R. Germain, P. Nietlisbach, C. Bousquet and S. Losdat for field support and C. Dawson (BC Ministry of Environment) for soil analyses. Field research complied with Canadian law. Comments by anonymous reviewers improved previous versions of the manuscript.

### Funding

Field research was supported by the Natural Sciences and Engineering Research Council of Canada, University of British Columbia and H. and W. Hesse. The funders had no role in study design, data collection and analysis, decision to publish, or preparation of the manuscript.

### Competing Interests

Louise K. Blight is an employee of Procellaria Research & Consulting, Victoria, British Columbia.

### Author Contributions

- Thomas K. Lameris conceived and designed the experiments, performed the experiments, analyzed the data, wrote the paper, prepared figures and/or tables, reviewed drafts of the paper.
- Joseph R. Bennett conceived and designed the experiments, performed the experiments, analyzed the data, wrote the paper, prepared figures and/or tables, reviewed drafts of the paper.
- Louise K. Blight reviewed drafts of the paper.
- Marissa Giesen performed the experiments, analyzed the data.
- Michael H. Janssen performed the experiments.
- Joop J.H.J. Schaminée conceived and designed the experiments.
- Peter Arcese conceived and designed the experiments, performed the experiments, wrote the paper, prepared figures and/or tables, reviewed drafts of the paper.

### Field Study Permissions

The following information was supplied relating to field study approvals (i.e., approving body and any reference numbers):

For permission to access study sites, we obtained permits from Parks Canada and the Gulf Islands National Park Reserve (permit number #GINP-2008-1433), the US Fish and Wildlife Service (permit number #07002) and were given oral permission by Tswaout and Tseycum bands and by A.J. Brumbaum and P. Green.

## Data Deposition

The raw data has been supplied as Supplemental Dataset Files.

## Supplemental Information

Supplemental information for this article can be found online at http://dx.doi.org/10.7717/peerj.2208#supplemental-information.

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
