# Peer review of "A century of ecosystem change: human and seabird impacts on plant species extirpation and invasion on islands"

_PeerJ, doi:10.7717/peerj.2208_

## Round 0.1 · original submission · Major Revisions

This is a valuable paper, as reflected by the reviewers' comments. However, they also raised important and constructive suggestions that you should address in your revision.

Reviewer 1 ·

Basic reporting

No comments.

Experimental design

No comments.

Validity of the findings

No comments.

Additional comments

Interesting article which i enjoyed reading. The link between humans and purported effects does not appear to be tested directly, but the authors provide a well-argued paper that human behaviour is ultimately responsible for the changes to the plant communities on the islands.

MINOR COMMENTS

line 170: "lme" has small "l"

lines 212-215: i think there is some problem here with reporting the results from Table 1. Please recheck these sentences.

line 273-274: too easy. what about nutrient inputs? these effects can persist for decades in some systems. if the alien plants thrive on high nutrient soils then they may continue to persist even if the physical disturbance caused by seabirds are removed.

Fig 1: Photos only from one location? not very convincing. can you provide additional locations? preferably a site with no bird colony?

Table 1: be consistent with the decimal places

Reviewer 2 ·

Basic reporting

This manuscript adheres to PeerJ's standards for basic reporting, however, titles should be added to the Supplementary Information. Currently, the SI tables do not appear to be labelled/captioned.

Experimental design

The data collection for recent decades is sufficient. The historical records are, however, limited although this is recognised by the authors (e.g. L223) and presumably unavoidable.

Validity of the findings

The findings appear valid and recognise the limited historical data.

Additional comments

Additional comments:
L75 “decimation” – this sounds non-scientific to me and may be misleading (the word is derived from Latin, meaning the removal of a tenth)
L76 amend italicised bracket
L140 “including” – error?
L145 unsure if the word “above” is needed given you've termed them the ‘gull islands’ previously
L148 -154 – consider improving the accessibility of this long sentence
L178 – “Supplementary material Appendix Table A1” – ensure this is consistent with titling of supplementary material.
L305 – error in sentence "Field research was complied with Canadian law"

Reviewer 3 ·

Basic reporting

The prose is excellent and apart from a handful of minor suggestions in the attached pdf I have no concerns with style etc.

That being said I do have some concerns with the inferences that are made, in part because there is insufficient background presented to convincingly support the conclusions reached.

These are outlined in the general comments to authors below.

Experimental design

The experimental design appears valid though I note the strongest conclusions in the manuscript are based on insight gained through qualitative approaches.

Validity of the findings

As stated above I have some concerns with the inferences that are made, in part because there is insufficient background presented to convincingly support the conclusions reached.

These are outlined in the general comments to authors below.

Additional comments

This is an interesting manuscript based mostly on qualitative observations but supplemented with a quantitative approach for vegetation data collected in 1986 and again in 2012.

Minor comments on the ms are attached as a marked up pdf file.

More substantial matters that should be addressed are outlined below:
• The case for human driven-seabird population change is not well made. Much of the manuscript is framed around this proposition but the reader is simply directed to other literature. If the current framework (human facilitated seabird population change drives vegetation change) is to be retained then this position needs to be more thoroughly articulated in the manuscript.
• The proposition that seabirds modify vegetation is a valid one but I think it is imperative that the authors present a convincing argument for why this is the principal plausible mechanism. I don’t believe other direct effects on vegetation are given consideration. This is especially important given the qualitative nature of many of the inputs. So for example the reader needs to be convinced that factors such as human visitation rate at these sites is not a suitable predictor of the responses that were observed. This is especially so given the cessation of aboriginal burning is implicated in historic vegetation changes – which in turn begs the question has anything else changed more recently that might also influence the observed vegetation shifts.
• Line 91 onwards. In the study system descriptors it would be valuable for the authors to describe key aspects of the islands that have the potential to influence establishment of exotic plants. How far is the island from the mainland (and sources of weeds)? What are human visitation rates like? Are there any restrictions on access to the island, including quarantine restrictions? Are there mammalian herbivores present? Do these differ for the other islands that do not have gulls?
• Line 106. The population trajectory of gulls breeding on the island is a key piece of information. I would like to see a plot showing the fluctuation in numbers over time to support the contention that numbers were low, then high, and have then been in decline since the 1980s.
• Line 274. It seems odd to predict that shrub cover might increase in the future given the decline in gulls commenced so long ago (“1980s”)….why is an increase in shrub cover not already occurring? If gulls have been in decline since the 1980s isn't it reasonable to expect such changes would already be detectable given that initial plant surveys were undertaken in 1986 and the most recent plant surveys were undertaken in 2012 (20-30 years after decline commenced)?? I can foresee reasons why this lag might exist but the onus is on the authors to highlight this mismatch and provide a robust explanation for the observation.

Annotated reviews are not available for download in order to protect the identity of reviewers who chose to remain anonymous.

---

## Round 0.2 · accepted · Accept

I agree with the reviewer that the manuscript has been improved and is now acceptable for publication.

Reviewer 1 ·

Basic reporting

The article is clearly written and mostly straightforward to follow.

Experimental design

Experimental design is sufficient given limitations on historical data.

Validity of the findings

The findings have now been presented with more careful interpretation.

Additional comments

The authors have addressed the concerns of both reviewers.